# Gray (*Oreochromis niloticus* x *O. aureus*) and Red (*Oreochromis* spp.) Tilapia Show Equal Susceptibility and Proinflammatory Cytokine Responses to Experimental Tilapia Lake Virus Infection

**DOI:** 10.3390/v11100893

**Published:** 2019-09-24

**Authors:** Kizito Kahoza Mugimba, Shlomit Tal, Saurabh Dubey, Stephen Mutoloki, Arnon Dishon, Øystein Evensen, Hetron M. Munang’andu

**Affiliations:** 1Department of Basic Sciences and Aquatic Medicine, Faculty of Veterinary Medicine, Norwegian University of Life Sciences, P.O. Box 369, Dep NO-0102, 1430 Ås Oslo, Norway; kahozak@gmail.com (K.K.M.); saurabh.dubey@nmbu.no (S.D.); stephen.mutoloki@nmbu.no (S.M.); oystein.evensen@nmbu.no (Ø.E.); 2Department of Biotechnical and Diagnostic Sciences, College of Veterinary Medicine Animal Resources and Biosecurity, Makerere University, P.O. Box 7062, Kampala, Uganda; 3Phibro Animal Health Corporation, R&D Vaccines, Ha’melacha St. 3, POB 489, West Industrial Zone, Beit-Shemesh 99100, Israel; shlomit.tal@gmail.com (S.T.); arnon.dishon@pahc.com (A.D.)

**Keywords:** tilapia lake virus, red tilapia, gray tilapia, IL-1β, TNFα

## Abstract

Tilapia is the second most farmed fish species after carp in the world. However, the production has come under threat due to emerging diseases such as tilapia lake virus (TiLV) that causes massive mortalities with high economic losses. It is largely unknown whether different tilapia strains are equally susceptible to TiLV infection. In the present study we compared the susceptibility of gray (*Oreochromis niloticus* x *O. aureus*) and red tilapia (*Oreochromis* spp.) to experimental TiLV infection. Virus was injected intraperitoneally at a concentration of 10^4^ TCID_50_/mL. Our findings show that gray tilapia had a lower mortality, 86.44%, but statistically not significantly different (*p* = 0.068) from red tilapia (100%). The duration of the mortality period from onset to cessation was similar for the two species, starting at 2–3 days post challenge (dpc) with a median at 10–11 dpi and ending on 20–22 dpi. In addition, there was no difference between species in mean viral loads in brain, liver and headkidney from fish collected soon after death. As for host response, expression levels of IL-1β and TNFα were equally high in brain and headkidney samples while levels in liver samples were low for both red and gray tilapia, which coincides with lower viral loads in liver compared to brain and headkidney for both species. We find that red and gray tilapia were equally susceptible to TiLV infection with similar post challenge mortality levels, equal virus concentration in target organs and similar proinflammatory cytokine responses in target and lymphoid organs at time of death. Nonetheless, we advocate that the search for less susceptible tilapia strains should continue with the view to reduce losses from TiLV infection in aquaculture.

## 1. Introduction

Tilapia is a common name for a group of cichlid fish that consist of close to 100 species. In the last decade tilapia has become the second most farmed fish species after carp worldwide [1]. Of the cultured tilapia, Nile tilapia (*Oreochromis niloticus*) accounts for the largest proportion (69.33%), followed by *Tilapia nei* (21.94%), Blue-Nile hybrid tilapia (*Oreochromis aureus* x *O. niloticus*) (7.80%), Mozambique tilapia (*Oreochromis mossambicus*) (0.01%) while the rest account for <0.01% [1]. However, the expansion of tilapia production has come under threat due to emerging diseases [2] with the most recent being tilapia lake virus (TiLV) having the potential to cause high economic losses [3,4]. TiLV is a negative sense single stranded RNA (−ssRNA) virus with 10 segments [3,5]. It is the only member in the genus *Tilapinevirus* in the *Amnoonviridae* family [6]. Since its first report in Israel in 2014, TiLV has been reported in various countries in Africa, Asia and South America where it has been associated with high mortalities [5,7,8,9,10,11]. However, there are reports of subclinical infections based on detection of TiLV nucleic acids while high mortalities associated with high replication of virus occur during the hot summer months [10,12,13].

Reports from TiLV affected farms show varying degrees of susceptibility among tilapiines [3,14,15,16]. Surachetpong et al. [9] reported TiLV massive outbreaks leading to varying mortality between 20–70% from 32 farms that involved Nile tilapia and red hybrid tilapia (*Oreochromis* spp.) in Thailand. They observed that mortalities were highest soon after transfer from hatcheries at grow-out stage in ponds and cages in the rivers. Similar outbreaks with varying mortality involving different tilapia strains have been reported in hot summer months in Israel [3]. Ferguson et al. [4] reported high mortality (80%) in a local on-farm genetically bred strain of Nile tilapia called “Chitralada” in Ecuador. On the contrary, another strain of Nile tilapia comprising of genetically bred males brought in from another producer remained uninfected. Similar to observations made by Surachetpong et al. [9], they noted that mortalities were highest soon after transfer from hatcheries to cages at the beginning of the grow-out stage. Tattiyapong et al. [14] reported lower mortality of 66% in red tilapia (*Oreochromis* spp.) compared to Nile tilapia (86%) experimentally injected by TiLV. Mugimba et al. [13] detected subclinical TiLV infections in farmed and wild Nile tilapia in Lake Victoria in Eastern Africa. Put together, these observations point to differences in susceptibility among tilapia strains. It still remains unknown whether there are underlying biological factors linked to differences in susceptibility among tilapia strains. A good understanding of differences in susceptibility among tilapia strains is vital for selective breeding of TiLV resistant strains for use in aquaculture.

The objective of this study was to determine whether gray and red tilapia are equally susceptible to TiLV based on post challenge mortality. In addition, we wanted to determine whether gray and red tilapia succumb to the same level of viral loads at the point of death as well as to determine whether they mount the same level of host responses based on proinflammatory cytokine responses. Hence, we compared the susceptibility of gray (*O. niloticus* x *O. aureus*) and red (*Oreochromis* spp.) hybrid tilapia to TiLV infection by comparing their post challenge survival proportions (PCSP) after intraperitoneal injection of the virus. We also examined if virus concentration in targets (liver and brain) and lymphoid (headkidney) organs was linked to mortality and compared findings in gray and red tilapia. Further, we compared the expression levels of the proinflammatory cytokines IL-1β and TNFα in the target and lymphoid organs of the two species as measures of host response to TiLV infection. Overall, we anticipate that data presented herein will contribute to understanding the differences to TiLV susceptibility among tilapiines.

## 2. Materials and Methods

### 2.1. Experimental Design

A total of 139 red tilapia hybrid (*Oreochromis* spp.) [4] and 142 gray tilapia hybrid (*O. niloticus* x *O. aureus*) weighing approximately 30 g, were used for the study. Six fish from each group (gray and red tilapia) were randomly selected and used for clinical examination followed by screening of various pathogens to ensure that fish used in the study were free of infections. The pathogens screened included TiLV, *Streptococcus agalactiae*, *S. iniae* and *Aeromonas hydrophila*. The remaining 133 red hybrid were divided into Tanks 1A and 1B with 43 and 30 fish for the TiLV challenge experiment, respectively, while the control red tilapia were divided into tanks 2A and 2B with each tank having 30 fish. Similarly, the remaining 136 gray tilapia for TiLV experimental challenge were divided into tanks 3A and 3B with 44 and 32 fish, respectively, while the control gray tilapia put in Tanks 4A and 4B were allocated 30 fish per tank. All fish used were obtained from Nir David Hatchery in Israel. All experiments were approved by the Norwegian Food Safety Authority (Mattilsynet FOTS ID: 18037 Date: 12 June 2019) After acclimatization for seven days in a re-circulation airflow system (RAS) at 28 °C, all used for experimental TiLV infection were intraperitoneally injected with 0.1 mL of virus at a concentration of 10^4^ TCID_50_/mL while all control fish were intraperitoneally injected with 0.1 mL phosphate buffered saline (PBS). The choice of challenge dose was based on a preliminary comparative study in which we found that the challenge dose of 10^4^ and 10^5^ TCID_50_/mL produced similar high mortality (>80%) while a challenge dose of 10^3^ TCID_50_/mL produced significantly low mortality (<60%). Therefore, 10^4^ TCID_50_/mL was chosen given its ability to produce high mortality (>80%) similar to ≥10^5^-TCID_50_/mL. Fish were fed ad libitum daily using commercial feed (Raanan fish feed, Israel). Fish were monitored daily for clinical signs and mortalities after TiLV challenge. All moribund fish were closely monitored, and all dying fish were collected soon after death. Liver, brain and headkidney samples were collected and stored in RNA later^®^ until use and no sample was collected for histopathology. Mortality was recorded daily for the determination of post-challenge survival proportions (PCSPs) using the Kaplan Meyer’s survival analysis.

### 2.2. RNA Extraction and cDNA Synthesis

Approximately 30 mg of liver, brain and headkidney tissues were homogenized in 1 mL Trizol reagent by bead beating with intermittent cooling on ice as previously described [17,18]. Homogenates were then centrifuged at 12,000× *g* for 10 min at 4 °C. Thereafter, the supernatant was transferred to new Eppendorf tubes and 0.2 mL chloroform was added followed by vortexing for 15 s. The mixture was incubated at room temperature for 5 min followed by centrifugation at 12,000× *g* for 15 min. Thereafter, the aqueous phase was transferred to new Eppendorf tubes followed by adding equal volume of freshly prepared 70% ethanol. After vortexing, the mixture was transferred to RNeasy spin columns. Thereafter, the Qiagen protocol was used based on the manufacturer’s guidelines (Qiagen, Hilden, Germany). RNA quality and quantification were measured using a Gen 5 3.0 microplate reader (BioTek Instrument Inc., Highland Park, Peabody, MA, USA) and gel electrophoresis analysis.

cDNA synthesis was carried out in 20 μL reaction volume using the Transcriptor First Strand cDNA Synthesis Kit (Roche diagnostics, Risch-Rotkreuz, Switzerland), which has a DNase treatment step, in two steps. Step one involved mixing 1 µg of template RNA with 2.5 µM Ancored-oligo (dT)_18_ primer and 60 µM random hexamer in a total volume of 13 µL. This mixture was treated with a denaturation step by heating at 65 °C for 10 min to remove secondary RNA structures followed by immediate cooling at 4 °C. The second step involved adding 8 nM MgCl_2_, 20 U of protector RNase inhibitor, 1 mM dNTPs and 10 U of transcriptor reverse transcriptase to each tube making final reaction volumes of 20 µL. The run profile in the second step was 25 °C for 10 min, 50 °C for 60 min and final reverse transcriptase inactivation step at 85 °C for 5 min. Reverse transcriptase has RNase H activity that removes RNA remnants improving downstream applications of the cDNA. This kit was selected because it uses Oligo-dT primers that have been reported to provide a reliable representation of the mRNA pool in the original RNA extract [19]. The final cDNA was stored at −20 °C until use for qRT-PCR assay.

### 2.3. Quantification of TiLV by Quantitative Real Time PCR in Tissue Samples

Virus quantification was carried out using cDNA synthesized from brain, liver and headkidney samples as described above. Primers targeting TiLV segment 3 (NCBI Genbank Acc No. KU552132) were designed using CLC workbench (Table 1). The LightCycler qPCR protocol composed of 95 °C denaturing step for 5 min, followed by 30 cycles of 95 °C for 10 s, 58 °C for 20 s and 72 °C for 10 s. This was followed by melting curve analysis as previously described [20]. Virus quantification was carried out by generating a linear standard curve using supernatants from TiLV infected TFC#10 cells whose concentration varied from 10^6^ TCID_50_/mL to 10^0^ TCID_50_/mL as previously described in our studies [21].

### 2.4. Gene Expression Analysis

Quantitative Real time PCR (qRT-PCR) was carried out using the SYBR^®^ green (Roche Applied BioSciences, Penzberg, Germany) detection method as previously described [17] to determine IL-1β and TNFα expression levels in the liver, brain and headkidney tissues from fish collected at 5, 9, 12 and 18 days soon after death during the mortality period. Primers for IL-1β and TNFα including the β-actin internal control genes were designed in the CLC workbench version 6 [22] (Table 1). qRT-PCR amplification cycles were carried out in a 96 Light-Cycler^®^ machine (Roche Applied BioSciences) while reactions were performed as described above using the following cycling conditions: 95 °C for 10 min initial denaturation; 95 °C for 3 s, annealing temperatures for 10 s, 72 °C for 30 s at 45 cycles. Melting curve analysis was done at 95 °C for 5 s followed by 65 °C for 1 min. Transcription levels for the target genes were quantified relative to the β-actin internal control genes using the delta–delta method as previously described [19].

### 2.5. Statistical Analysis

The Kaplan Meyer’s survival analysis was used to compute the proportion of fish that survived the TiLV challenge using mortality data recorded on a daily basis in GraphPad Prism^®^ versions 5.0, while the Cox hazard risk analysis was used to compare the risk of TiLV infected gray and red tilapia relative to the uninfected control fish. The Cq mean values obtained from qRT-PCR described above were used to calculate log_2_ fold change of IL-1β and TNFα as previously described using β-actin as the internal control housekeeping gene [23]. Data analysis was performed in Prism version 5.0.

## 3. Results

### 3.1. Clinical Observations and Kaplan Meyer’s Survival Analysis

Post challenge clinical observations mainly comprised of irregular swimming behavior starting at 2 days post challenge (dpc). Gross pathological changes were characterized by hemorrhages on skin surfaces especially around the gill operculum and basal fin area. Some fish showed signs of exophthalmia. Survival analysis showed end-point survival of 13.56% and 0% for grey and red tilapia, respectively. The onset of mortality for gray and red tilapia started at 2 and 3 dpc (Figure 1), respectively. The median of 50% mortality for gray and red tilapia was at 10 and 11 dpi (Table 2), respectively, and mortality continued up to 20 dpi. Red and gray tilapia had the same duration of mortality from onset (2–3 dpi), through median mortality (11–12 dpi) ending at 20–22 dpi (Figure 1). There was no significant difference (*p* = 0.068) in mortality between red and gray tilapia. Table 2 shows that the risk of red tilapia (19.51) dying due to TiLV was slightly higher than gray tilapia (16.98).

### 3.2. Comparison of Virus Concentration in Brain, Headkidney and Liver Samples

Figure 2 shows quantification of TiLV in brain, headkidney and liver tissues of gray and red tilapia collected at four different time-points over the course of the challenge period. The mean virus concentration for the brain, headkidney and liver samples was estimated at 6.188, 6.103 and 5.871 log_10_ TCID_50_/mL for red tilapia, respectively. The mean virus concentration for the liver was significantly lower (*p* = 0.0361) than the mean virus quantity for brain samples for red tilapia. However, there was no significant difference (*p* > 0.9999) in the mean virus concentration between brain and headkidney, as well as between headkidney and liver samples (*p* = 0.6536) for red tilapia. The mean virus concentration for the brain, headkidney and liver samples for gray tilapia was estimated at 6.164, 6.112 and 5.872 log_10_ TCID_50_/mL, respectively. The mean virus concentration (over the course of the challenge period) for the liver was significantly lower (*p* = 0.048) than in brain samples, but there was no significant difference (*p* > 0.9999) between the brain and headkidney as well as between the liver and headkidney samples (*p* = 0.0565) for gray tilapia.

### 3.3. Gene Expression

Figure 3 shows a comparison of IL-1β expression levels in brain, headkidney and liver samples collected from red and gray tilapia at four different time-points over the challenge period. Generally, highest expression levels were in the brain followed by the headkidney. However, IL-1β expression levels were significantly lower in the liver than in the brain and headkidney samples for both red and gray tilapia. The mean Log_2_ fold change for IL-1β for red tilapia was estimated at 3.615, 1.844, and 0.642 for brain, headkidney and liver samples, respectively, while the mean Log_2_ fold change for IL-1β for gray tilapia was estimated at 3.903, 1.921 and 1.072, for the brain, headkidney and liver for gray tilapia, respectively. As for comparison between red and gray tilapia, there was no significant difference (*p* = 0.513) in levels of IL-1β observed in brain samples between red and gray tilapia and no difference was observed in headkidney samples (*p* = 0.840) as well as in liver samples (*p* = 0.140) between the two tilapia strains.

Figure 4 shows expression levels of TNFα for brain, headkidney and liver for red and gray tilapia. The mean log_2_ fold change in levels of TNFα in the brain, headkidney and liver samples for red tilapia were 2.729, 1.529 and −0.326, respectively, while the mean Log_2_ fold change for TNFα for gray tilapia was estimated at 3.265, 1.164 and 0.122 for the brain, headkidney and liver, respectively. There was no significant difference observed in expression levels of TNFα in brain (*p* = 0.143), headkidney (*p* = 0.507) or liver samples (*p* = 0.358) between red and gray tilapia.

## 4. Discussion

The main conclusion to be drawn from this study is that red and gray tilapia are equally susceptible to experimental TiLV infection. This aligns with similar viral titers in target organs and aligned cytokine responses post challenge. There was a higher end-point mortality in red tilapia but not statistically different from grey tilapia. Moreover, the dynamics of the mortality curve was similar for the two species with mortality starting at 2–3 dpi with a median of 10–11 dpi and cessation at 20–22 dpi. Cox hazard risk analysis showed no difference in risk of gray and red tilapia dying due to TiLV infection. These findings are in line with observations made for Nile tilapia, which is the most commercially used tilapia strain [1], in which high mortality (>80%) has been widely reported from various countries such as Peru, Ecuador and Thailand [4,9,11].

The brain and liver are considered the target organs for TiLV and liver damage has been linked to syncytial formation in hepatocytes, while brain pathology has been linked to irregular swimming seen in infected fish, consistent with clinical observations in this study [13,24]. As for comparison between red and gray tilapia, our findings show that there was no significance difference (*p* > 0.9999) in the mean virus concentration from brain samples obtained from the two tilapia strains as shown from 6.188 and 6.164 log_10_ TCID_50_/mL mean viral loads for red and gray tilapia, respectively. Similarly, there was no significance difference (*p* > 0.9999) in the mean virus concentration from headkidney samples as shown from the 6.103 and 6.112 log_10_ TCID_50_/mL mean viral loads for red and gray tilapia, respectively. This trend was also consistent for liver tissues (*p* > 0.9999) in which red and gray tilapia had mean viral loads of 5.871 and 5.872 log_10_ TCID_50_/mL, respectively. Put together, these findings point to equal permissiveness of TiLV replication in these organs. For some diseases, it has been shown that 10^7^ TCID_50_/mL viral loads in target organs correspond with establishment of pathology and high mortality [25,26]. Although we did not assess pathology in target organs, it is likely that the high viral loads of 6.188, 6.103 and 5.871 log_10_ TCID_50_/mL for red tilapia and 6.164, 6.112 and 5.872 log_10_ for gray tilapia detected in brain, headkidney and liver samples could be linked to pathology. Further studies should be carried out to link viral load with pathology, as well as linking viral loads with mortality.

The viral loads in headkidney of both species are high, 10^6^ TCID_50_/mL of tissue, and if this constitute a primary replication site for TiLV or drainage into lymphoid organs need to be better understood. There are also few studies addressing the pathology in headkidney during TiLV infection and additional studies would be required to better understand the role that headkidney play in the propagation of the infection. These observations imply that both red and gray tilapia succumb to similar viral loads in target and lymphoid organs at the point of death, consolidating our view that they are equally susceptible to TiLV infection.

Host responses play a role in protection and propagation of the infection/tissue damage as part of the infection. IL-1β and TNFα expression levels can be used to better understand the host response to infection and expression of IL-1β and TNFα in response to various infections has been reported previously in tilapia [27,28]. We found that the brain had the highest levels of IL-1β and TNFα expression followed by the headkidney while levels in the liver were low for both species. Both red and gray tilapia had comparable IL-1β and TNFα levels in the brain and headkidney and expression levels were similar at different points, which corresponded with equal viral loads across time points post challenge. The number of cytokines sequenced for tilapia is limited and other markers of viral infection, like interferon alpha and possibly gamma with down-stream effectors (Mx), would likely provide additional insight into the host responses to viral infection. However, up until now, these sequences are not available and future studies should include more detailed assessment of viral response genes when they become available, with an aim to better understand elements of host responses that fail to prevent viral progression.

In summary, we have shown that gray and red tilapia are equally susceptible to TiLV based on PCSP and virus quantification in target and lymphoid organs. In addition, the host response to TiLV infection, TNF-α and IL1-β mRNA expression was also similar. We advocate that the search for less susceptible tilapia strains to TiLV infection should continue with the view to finding resistant strains against this highly pathogenic virus.

## Figures and Tables

**Figure 1 viruses-11-00893-f001:**
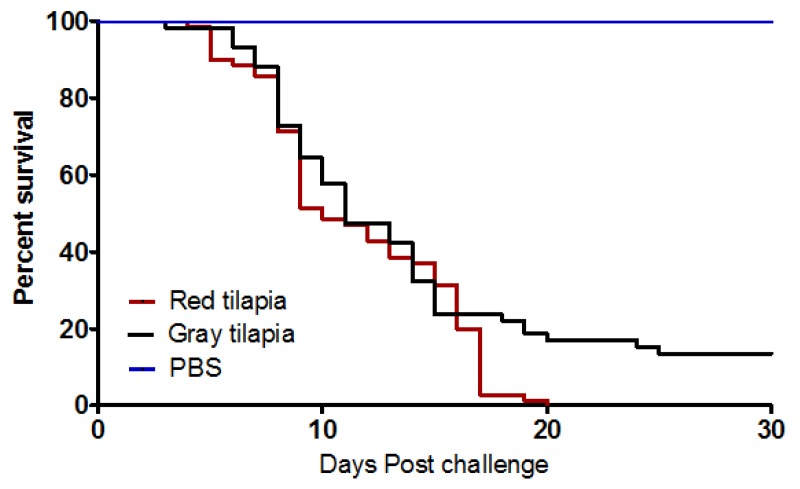
Kaplan Meyer’s survival analysis comparing the post challenge survival proportions of gray and red tilapia injected by 1 × 10^4^ TCID_50_/mL of tilapia lake virus (TiLV) intraperitoneally.

**Figure 2 viruses-11-00893-f002:**
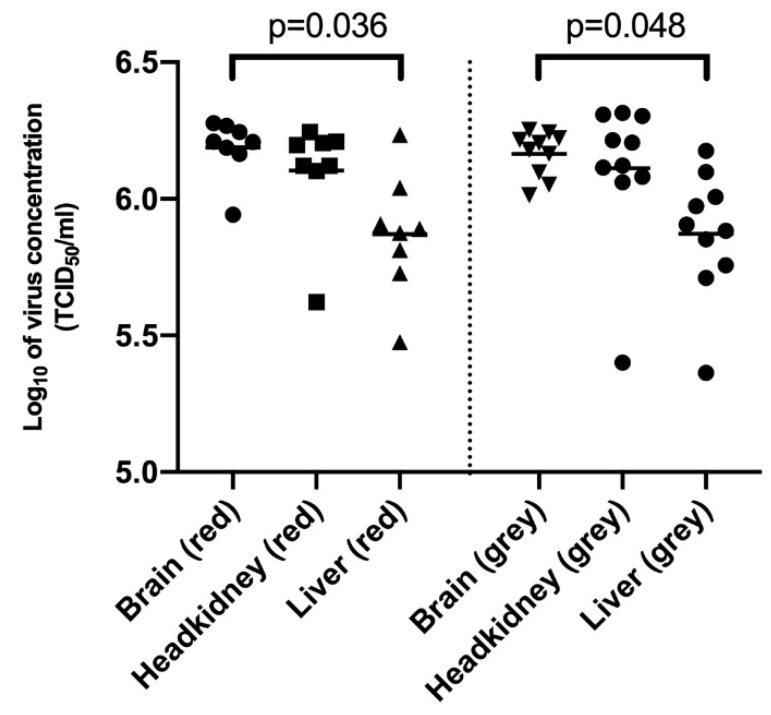
Comparison of virus concentration in the liver, brain and headkidney samples of gray tilapia (right panel) and red tilapia (left panel) compiled over the mortality period.

**Figure 3 viruses-11-00893-f003:**
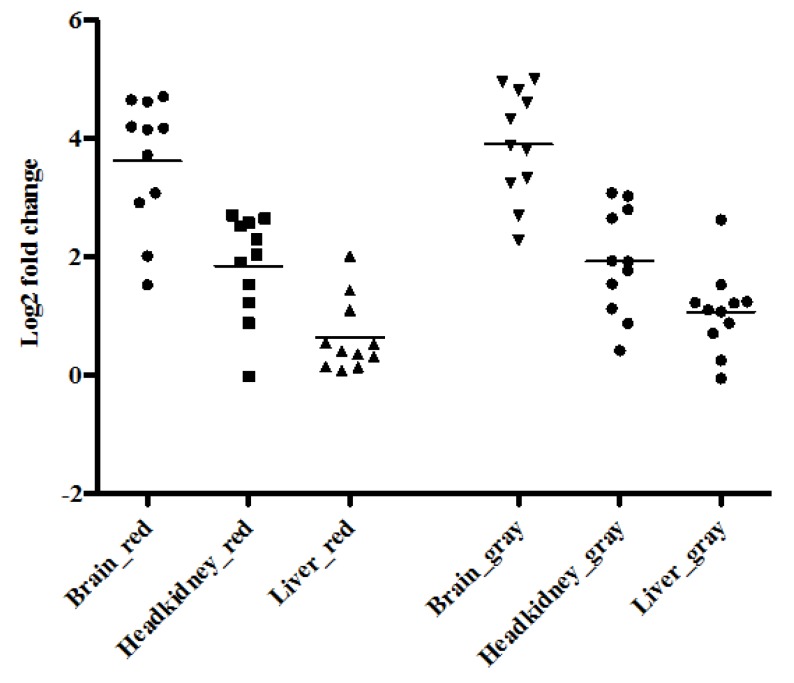
Expression levels of IL1-β in brain, headkidney and liver samples of red and gray tilapia compiled over the mortality period.

**Figure 4 viruses-11-00893-f004:**
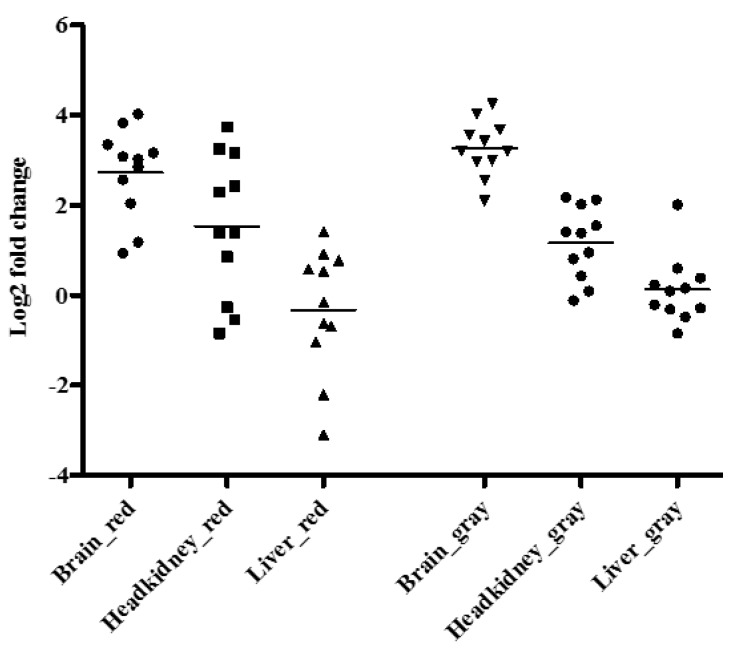
TNFα expression in the brain, headkidney and liver of gray and red tilapia compiled over the mortality period.

**Table 1 viruses-11-00893-t001:** Primer sequences used for virus quantifications and gene expression.

Primer ID	NCBI		Primer Sequence	Bp Length	Tm (°C)
IL-1β	KF747686.1	F	TGGAGGAGGTGACGGATAAA	86 bp	62 °C
		R	GGTGTCGCGTTTGTAGAAGA		
TNFα	NM_001115056	F	GGCTAGATTTCCTCTGCTGTATC	79 bp	62 °C
		R	GCTATGACAGCACCTCTGTATC		
β-actin	KJ126772.1	F	GTGGGTATGGGTCAGAAAGAC	111 bp	62 °C
		R	GTCATCCCAGTTGGTCACAATA		
TiLV Seg 3	KU751816	F	TCCAGATCACCCTTCCTACTT	109 bp	62 °C
		R	ATCCCAAGCAATCGGCTAAT		

**Table 2 viruses-11-00893-t002:** Cox hazard proportion risk analysis.

Parameters	Fish breed/Strain	
	Gray Tilapia	Red Tilapia
Controls	60	60
Number of fish tested	76	73
Number of tanks	2	2
Post challenge Mortality %	80.4%	100.0%
Hazard risk ratio	16.98	19.51
95% Conf Interv Hazard risk	09.40–30.68	11.62–32.96
Media survival	10	11

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
