# Peer review of "Gray (Oreochromis niloticus x O. aureus) and Red (Oreochromis spp.) Tilapia Show Equal Susceptibility and Proinflammatory Cytokine Responses to Experimental Tilapia Lake Virus Infection"

_viruses, 2019, doi:10.3390/v11100893_

Round 1

Reviewer 1 Report

The present manuscript entitled ‘Gray (Oreochromis niloticus x O.aureus) and red (Oreochromis spp.) tilapia show equal susceptibility and proinflammatory cytokine responses to experimental tilapia lake virus infection’ reports the mortality, virus titres (in brain, liver and head kidney) and pro-inflammatory cytokine levels at the transcript level (both IL-1beta and TNF-alpha) in TiLV infected gray and red tilapia. This manuscript is well written, and though the findings aren’t extensive, they are well executed and presented. There are some modifications required prior to publication.

Major:

Please justify why virus load and cytokine production was only measured in dead fish. It seems to be there would be value in looking at virus titres in live vs dead fish at different time points. Additionally, there would be value in looking at cytokine levels in live vs dead fish at different time points. Please justify why cytokine levels were measured in the dead fish only. What happened in the gray tilapia fish that survived the virus challenge? Were cytokine levels elevated in these fish? The decision to test only dead fish needs to be explained and defended for both virus titre and cytokine production experiments.

Minor:

Please explain in the manuscript why the titre of 104TCID50/fish was chosen for this trial. Please include in the discussion a few sentences to compare what has been found in this manuscript with that has been demonstrated in more commercially valuable species, such as Nile Tilapia. Line 39: include the genus species for Blue-Nile hybrid tilapia Line 45: change ‘African’ to ‘Africa’ Line 79: include the names of the pathogens against which the fish were screened. Line 91: Define the abbreviation ‘PCSPs’ Line 103: was the RNA DNase treated? If not, explain what precautions were taken to ensure only cDNA and not contaminating gDNA was detected in the samples. Please ensure consistency with head kidney. There are a number of places where it is referred to as kidney. (Lines: 116, 130, 170). Line 264: please change ‘till’ to ‘until’

Author Response

QUERY-1: Please justify why virus load and cytokine production was only measured in dead fish. It seems to be there would be value in looking at virus titres in live vs dead fish at different time points. Additionally, there would be value in looking at cytokine levels in live vs dead fish at different time points. Please justify why cytokine levels were measured in the dead fish only. What happened in the gray tilapia fish that survived the virus challenge? Were cytokine levels elevated in these fish? The decision to test only dead fish needs to be explained and defended for both virus titre and cytokine production experiments.

RESPONSE: The main aim of this study was to determine whether red and gray tilapia succumb to the same level of mortality as well as the same of quantity of virus at the point of death and also to determine whether they express the level of host responses based on proinflammatory cytokine respones. We certainly want to proceed with a follow-up study where we will compare the sequential progression of infection and collect samples during the incubation, acute and convalescent stages after TiLV infection. But for the current study our focus was to compare susceptibility based on mortality, virus quantification and gene expression only at the point of death. This is now explained in the manuscript in the objectives part, see line 65-68.

QUERY-2: Please explain in the manuscript why the titre of 104TCID50/fish was chosen for this trial.

RESPONSE: This was based on a preliminary study carried out in a previous study, not yet published. But we have explained in the manuscript (See line 90-94)

QUERY-3: Please include in the discussion a few sentences to compare what has been found in this manuscript with that has been demonstrated in more commercially valuable species, such as Nile Tilapia.

RESPONSE: We have added a few sentences in the manuscript explaining this part and we have included some references (see lines 246-248)

QUERY-4: Line 39: include the genus species for Blue-Nile hybrid tilapia. Line 45: change ‘African’ to ‘Africa’

RESPONSE: We have included the genus name for blue-Nile tilapia, and the word ‘African’ has been changed to ‘Africa’ as recommended, see line 39 and 45.

QUERY-5: Line 79: include the names of the pathogens against which the fish were screened.

RESPONSE: Names of pathogens have been included, see line 83

QUERY-6: Line 91: Define the abbreviation ‘PCSPs’

RESPONSE: PCSPs has been defined as ‘Post challenge survival proportions’ see line 99-100

QUERY-7: Line 103: was the RNA DNase treated? If not, explain what precautions were taken to ensure only cDNA and not contaminating gDNA was detected in the samples.

RESPONSE: The DNAse treatment step is included in the Roche Kit used in the study, this is now mentioned in the manuscript, see line 113.

QUERY-8: Please ensure consistency with head kidney. There are a number of places where it is referred to as kidney. (Lines: 116, 130, 170).

RESPONSE : ‘Kidney’ has been replaced with one word ‘headkidney’ throughout the manuscript, see lines 125, 139 and 178.

QUERY-9: Line 264: please change ‘till’ to ‘until’

RESPONSE: Correction done as recommended.

Reviewer 2 Report

Gray (Oreochromis niloticus x O. aureus) and red (Oreochromis spp.) tilapia show equal susceptibility and proinflammatory cytokine responses to experimental tilapia lake virus infection

By Mugimba et al.

viruses-583705

This manuscript represents an interesting contribution to the knowledge of fish viruses and their specific effects. The study is rather simple but proposes various techniques to assess such specificity from survival to a gene expression. There are a few points that would need more clarifications and an additional proofcheck but overall, this paper could be published pending minor modifications. For instance, we need to know where are the fish used in the experiment from. Also, how many control fish did you use from the total (l.85)? I would also like to see the analysis of the different time points before they are pooled (Fig. 3 and 4). Gene expression is often time-dependent so this information would be very interesting here.

Here are some editorial suggestions:

- line 22 and 156: “the duration of mortality”, do you mean the time to death?

- line 31: I am not familiar with the fish semantic but is “strains” the right term for tilapia

- line 48 and 73: tilapines or tilapiines?

- line 49, 58 and others: reported TiLV… reported mortality.. Reported is a direct verb, no need for a preposition

- line 59: compared to Nile tilapia (86%)

- line 80 and following: replace allocated by with?

- line 87: Israel).

- line 130: dead or death?

- line 142-143: tilapia relative to

- Table 2: This table should be consistent with the numbers; as it is we can not find the groups or 30 fish nor the controls  

- line 179: the p-value should come after significantly lower

- Figure 2: the rounded p-value on panel B should be 0.049

- line 235: two species, with mortality

Best regards,

Author Response

QUERY-1: For instance, we need to know where are the fish used in the experiment from. Also, how many control fish did you use from the total (l.85)?

RESPONSE: The name of the supplier of fish used in the study is now mentioned in the manuscript (see line 88) and the number of control fish is well explained in the current submission. We have re-written this part to make it more simple and easier to understand (see line 79-87)

QUERY-2: I would also like to see the analysis of the different time points before they are pooled (Fig. 3 and 4). Gene expression is often time-dependent so this information would be very interesting here.

RESPONSE: While we do understand that gene expression is time depend, in our view this would have been suitable for studying fish response from challenge until death while the present only focused in gene expression at the time of death. Certainly gene expression is a ‘time dependent factor’ and in our view it is good to study gene expression as a ‘time dependent factor’ alongside ‘infection progression’ so that gene expression is used to determine host response at different stages of infection progression. But in this case, the comparison between red and gray tilapia was focused at the point of death. The point was to compare whether the two strains had the same level of mortality, quantity of virus and level of proinflammatory response at the time of death. We do hope the reviewer will be considerate in understanding our approach in the current study.

QUERY-4: - line 22 and 156: “the duration of mortality”, do you mean the time to death?

RESPONSE. We mean from onset to cessation of mortality, this has been explained in the manuscript, see lines 23 and 164.

QUERY-5: - line 31: I am not familiar with the fish semantic but is “strains” the right term for tilapia

RESPONSE. Consulting with fish experts/breeders, we were advised that word ‘strain’ is fine.

QUERY-6: - line 48 and 73: tilapines or tilapiines?

RESPONSE: Correction done, tilapiines is correct (see line 48 and 76)

QUERY-7: - line 49, 58 and others: reported TiLV… reported mortality.. Reported is a direct verb, no need for a preposition

RESPONSE: Correction done as recommended, see line 49 and 58.

QUERY-8: - line 59: compared to Nile tilapia (86%)

RESPONSE: Correction done as recommended, see line 59

QUERY-9: - line 80 and following: replace allocated by with?

RESPONSE: Correction done as recommended, see line 84

QUERY-10: - line 87: Israel).

RESPONSE: Correction done as recommended, see line 96

QUERY-11: - line 130: dead or death?

RESPONSE: Changed to death as recommended, see line 140

QUERY-12: - line 142-143: tilapia relative to

RESPONSE: Changes done as recommended, see line 151-152

QUERY-13: - Table 2: This table should be consistent with the numbers; as it is we can not find the groups or 30 fish nor the controls  

RESPONSE: Table 2 has now been harmonized with the numbers in the manuscript (see Lines 79-87 and Table 2)

QUERY-14: - line 179: the p-value should come after significantly lower

RESPONSE: Changes done as recommended, see line 187.

QUERY-15: - Figure 2: the rounded p-value on panel B should be 0.049

RESPONSE: Changes done as recommended 187

QUERY-16: - line 235: two species, with mortality

RESPONSE: Changes done as recommended 244

Reviewer 3 Report

76 to 77 Experimental design. I think that the total number of fish were 60 control+ 79 inoculated for red tilapia, and 60 contol + 82 inoculated for gray tilapia. Is this correct? Then, they should be a total of 139 red tilapia hybrid, and 142 gray tilapia

101 Revise “QiaGen, Germany”

112 Revise “Olgi-dT Primers”,  Could be “Oligo-dT primers”?

115. Virus quantification. The titre of the virus should be determined by cell culture infection (E-11 cell line, for example). It is possible that the same amount of RNA gives different amount of final viable virus.

120 to 121. The redaction of melting curve is confused. It could be something similar to this (it is a suggestion): “To determine the amplified product melting temperature, after SYBR Green qPCR, the temperature was raised from XX to XX °C and the fluorescence detected during 10 s after each 0.X °C increase”.

Author Response

QUERY-1: 76 to 77 Experimental design. I think that the total number of fish were 60 control+ 79 inoculated for red tilapia, and 60 contol + 82 inoculated for gray tilapia. Is this correct? Then, they should be a total of 139 red tilapia hybrid, and 142 gray tilapia

RESPONSE: Agreed and this part is well explained in the current submission, see first paragraph in the methods and Materials Section

QUERY-2: 101 Revise “QiaGen, Germany”

RESPONSE: Changes done as recommended, see line 110

QUERY-3: 112 Revise “Olgi-dT Primers”,  Could be “Oligo-dT primers”?

RESPONSE: Changes done as recommended, see line 122

QUERY-4: 115. Virus quantification. The titre of the virus should be determined by cell culture infection (E-11 cell line, for example). It is possible that the same amount of RNA gives different amount of final viable virus.

RESPONSE: The virus quantity was determined by standard curve and this is mentioned in line 130-133.

QUERY-5: 120 to 121. The redaction of melting curve is confused. It could be something similar to this (it is a suggestion): “To determine the amplified product melting temperature, after SYBR Green qPCR, the temperature was raised from XX to XX °C and the fluorescence detected during 10 s after each 0.X °C increase”.

RESPONSE: Correction has been done by referring readers to previous work where the same method was used and reference has been provided.
